# Postmastectomy Radiation Therapy (PMRT) before and after 2-Stage Expander-Implant Breast Reconstruction: A Systematic Review

**DOI:** 10.3390/medicina55060226

**Published:** 2019-05-29

**Authors:** Jeremie D. Oliver, Daniel Boczar, Maria T. Huayllani, David J. Restrepo, Andrea Sisti, Oscar J. Manrique, Peter Niclas Broer, Sarah McLaughlin, Brian D. Rinker, Antonio Jorge Forte

**Affiliations:** 1Mayo Clinic Alix School of Medicine, Mayo Clinic, Rochester, MN 55905, USA; oliver.jeremie@mayo.edu; 2Division of Plastic Surgery and Robert D. and Patricia E. Kern Center for the Science of Health Care Delivery, Mayo Clinic, Jacksonville, FL 32224, USA; danielboczar92@gmail.com (D.B.); maria.t.huayllanip@gmail.com (M.T.H.); rpo20@hotmail.com (D.J.R.); asisti6@gmail.com (A.S.); rinker.brian@mayo.edu (B.D.R.); 3Division of Plastic Surgery, Mayo Clinic, Rochester, MN 55905, USA; manrique.oscar@mayo.edu; 4Department of Plastic, Reconstructive, Hand and Burn Surgery, Bogenhausen Academic Hospital, 81925 Munich, Germany; niclas.broer@klinikum-muenchen.de; 5Division of General Surgery, Mayo Clinic, Jacksonville, FL 32224 USA; mclaughlin.sarah@mayo.edu

**Keywords:** postmastectomy radiation therapy, PMRT, implant-based reconstruction, breast implant, radiation, breast cancer, breast reconstruction, infection, explantation

## Abstract

*Background*: In those undergoing treatment for breast cancer, evidence has demonstrated a significant improvement in survival, and a reduction in the risk of local recurrence in patients who undergo postmastectomy radiation therapy (PMRT). There is uncertainty about the optimal timing of PMRT, whether it should be before or after tissue expander or permanent implant placement. This study aimed to summarize the data reported in the literature on the effect of the timing of PMRT, both preceding and following 2-stage expander-implant breast reconstruction (IBR), and to statistically analyze the impact of timing on infection rates and the need for explantation. *Methods*: A comprehensive systematic review of the literature was conducted using the PubMed/Medline, Ovid, and Cochrane databases without timeframe limitations. Articles included in the analysis were those reporting outcomes data of PMRT in IBR published from 2009 to 2017. Chi-square statistical analysis was performed to compare infection and explantation rates between the two subgroups at *p* < 0.05. *Results*: A total of 11 studies met the inclusion criteria for this study. These studies reported outcomes data for 1565 total 2-stage expander-IBR procedures, where PMRT was used (1145 before, and 420 after, implant placement). There was a statistically significant higher likelihood of infection following pre-implant placement PMRT (21.03%, *p* = 0.000079), compared to PMRT after implant placement (9.69%). There was no difference in the rate of explantation between pre-implant placement PMRT (12.93%) and postimplant placement PMRT (11.43%). *Conclusion*: This study suggests that patients receiving PMRT before implant placement in 2-stage expander–implant based reconstruction may have a higher risk of developing an infection.

## 1. Introduction

In the United States, the most common types of cancer (and most common causes of cancer death) in women include lung, colorectal, and breast, representing conjointly one-half of all cases reported [1]. Of these, 30% of all new cancer diagnoses in women are isolated breast cancer [1]. An estimated 266,120 women were diagnosed with breast cancer in the United States in 2018 [1]. According to the American Society of Plastic Surgeons, 109,256 breast reconstruction procedures were performed in 2016; this number represents an increase in 39% compared to that in 2000 [2]. Implant-based breast reconstruction (IBR) is the most prevalent technique [3,4].

The National Cancer Database showed that between 2014 and 2015, 305,391 women under 70 years of age underwent surgery as a treatment for breast cancer in the United States [5,6]. The most common breast reconstruction technique to date consists of using a 2-stage tissue expander/implant [7]. Considering that 35% to 40% of female patients diagnosed with breast cancer every year undergo a mastectomy, it is interesting that less than 25% undergo immediate reconstruction [8]. The choice to perform postmastectomy radiation therapy (PMRT) is a complex decision that requires careful analysis of many factors [9,10]. Currently accepted indications for PMRT include four or more positive axillary lymph nodes, primary tumor size greater than 5 cm, and operable stage III breast cancer [11].

Evidence has demonstrated substantial survival improvement and a reduction in the risk of local recurrence in patients who undergo PMRT [7,9]. Nevertheless, approximately 95% of patients treated with radiation therapy develop acute radiation dermatitis, which is an acute inflammatory reaction consisting of erythema, swelling, edema, desquamation and ulceration [12]. Furthermore, irradiated tissue releases transforming growth factor beta, inducing fibrocyte proliferation and leading to chronic tissue changes (e.g., radiation-induced fibrosis and atrophy on the skin and subcutaneous tissue of the breast, which consists of skin retraction, discoloration, induration and decreased treated breast volume) [13,14].

The necessity and utility of PMRT have the potential to complicate outcomes in breast reconstruction substantially. Precautions are mandated to mitigate these potential risks by adopting a multidisciplinary, patient-centered approach to care, in which the following potential risks of PMRT are fully explained to the patient: Potential toxicities (short- and long-term), mortality prognosis, benefits of therapy, tumor characteristics, total tumor burden, and future plans for systemic treatment [15]. This comprehensive systematic review aimed to summarize the data reported in the literature on the effect of the timing of PMRT, both preceding and following IBR, and to statistically analyze respective rates of infection and explant between preimplant and postimplant PMRT subgroups.

## 2. Methods

A comprehensive systematic review of the literature was conducted using the PubMed/Medline, Ovid, and Cochrane databases without timeframe limitations. Articles included in the analysis were those reporting outcomes data of PMRT in implant breast reconstruction (IBR), published from 2009 to 2017. The following Medical Subject Headings (MeSH) were used: “radiation breast reconstruction” and “breast implant and radiation”, with the following keywords applied and scrambled in all combinations: “breast reconstruction”, “postmastectomy radiation therapy”, “radiation therapy”, “tissue expander/implant”, “breast conservation therapy”, “reconstructive failure”, “capsular contraction”, “breast implant”, “breast cancer”, and “postmastectomy radiotherapy”.

The compiled reference lists were compared and reviewed for potential relevance. The bibliographies of included studies were also searched for missed articles. After all of the authors had completed their systematic literature review, additional verification through the Mayo Clinic Library (Rochester, MN, USA) was requested and received, to ensure the most comprehensive review of all published studies meeting these criteria was performed. Preclinical studies, non-English text, abstract only, and articles failing to stratify outcomes based on radiation timing were excluded. This study followed the PRISMA guidelines for article identification, exclusion, and final selection.

## 3. Results

After applying inclusion and exclusion criteria to all related articles, a total of 11 studies met the criteria for this study. These studies reported outcomes data for 1565 total 2-stage expander-IBR procedures wherein PMRT was used (1145 before and 420 after initiation of 2-stage expander-implant placement) (Table 1). There was a statistically significant higher likelihood of infection following pre-implant placement PMRT (21.03%, X^2^ = 15.5813, *p* = 0.000079) compared to PMRT after implant placement (9.69%) (Table 2). There was no difference in the rate of explantation between pre-implant placement PMRT (12.93%) and post-implant placement PMRT (11.43%, X^2^ = 0.6287, *p* = 0.427821) (Table 3).

## 4. Discussion

It has been largely demonstrated that administering radiation therapy prior to breast reconstruction increases complication rates with tissue expander and IBRs [16,17,18,19]. Brooks et al. [16] observed a higher likelihood of successful tissue expander plus implant reconstruction in irradiated patients younger than 50 years with a body mass index (BMI) < 30. IBR following PMRT has shown a higher incidence of complications compared to staged autologous reconstruction; nevertheless, it remains the favored reconstructive option [20].

Radiation therapy has become more common in breast cancer management as a result of improvements in technology and increased evidence of benefits; however, it poses a challenge for plastic surgeons, who need to consider the optimal timing and method for breast reconstruction in these patients [20,21]. Several studies have demonstrated that radiotherapy following prosthetic reconstruction is associated with a higher risk of reconstruction failure, overall complications, and capsular contracture compared to reconstruction without radiotherapy [7,22,23,24,25,26,27,28,29,30].

Two large series of prospective immediate 2-stage breast reconstructions reported a failure rate of 9.1% in patients who underwent PMRT after immediate IBR, compared to only 0.5% in patients who underwent breast implant reconstruction without undergoing radiation. Current data on PMRT and its effect on the breast capsule has been inconclusive, warranting further studies on this topic [15,31].

The pathogenesis of capsular contracture is not well understood. It is a multifactorial process that involves human body reaction, biofilm activation, and bacteremic seeding or silicon exposure [32]. Silicon prostheses have shown more structural, mechanical, and chemical modifications secondary to radiotherapy than polyurethane implants [32]. Direct-to-implant reconstruction has shown fewer failure rates with implants than expander reconstruction in patients undergoing PMRT [33,34,35,36].

A consensus has not been reached on what is the optimal timing to exchange tissue expanders with a breast implant in patients undergoing 2-stage expander-implant reconstruction [4,37]. While the exact timing (weeks/months) of PMRT before or after expander-implant reconstruction was not always reported in the studies included in this review, there was a range of timing in those that did report. Multi-center prospective trials are warranted to elucidate the proper timing for PMRT in regards to patient outcomes.

Patients reported a highly variable failure rate of reconstruction (0–40%), which seemed to depend on whether radiotherapy was delivered, either to the tissue expander or the implant [37]. In 2011, Nava and colleagues [36] published a study reporting that 20 (40.0%) of 50 patients experienced failed implant-based reconstruction when radiotherapy was delivered to the tissue expander, compared to only 7 (6.4%) of 109 patients who were treated with radiotherapy to only the implant (*p* < 0.0001). Based on the surgeons’ assessment of the reconstructed breast shape and symmetry, a higher number of positive results is seen in patients who receive radiation therapy to placed implants, than in those only with a tissue expander [36]. Capsular contracture (Baker grade IV) was shown to be more common in patients who received PMRT on tissue expander than in patients who received PMRT on breast implants [36,37]. 

Furthermore, a study by Cordeiro et al. [34] reported a higher incidence of breast reconstruction failure in patients who received radiotherapy with a tissue expander than patients with breast implants, though this finding was not statistically significant (18.1% vs. 12.4%). No substantial differences between both radiotherapy groups have been reported in other retrospective datasets [37,38,39].

In 2017, a meta-analysis that analyzed data from 899 cases (489 received radiotherapy to the tissue expander and 410 to the permanent implant) showed that the risk of reconstruction failure was not significantly higher in patients who received radiotherapy to the tissue expander than to the implant [40].

Factors such as dosage, treatment length, and the timeline of therapy all contribute to the extent to which PMRT ultimately impacts implant outcomes [41,42,43]. A shortcoming in the current literature is that few of these risk factors have been clinically quantified, and therapy guidelines are still lacking. In light of this, many centers depend upon expert opinion or preference, patient choice, and isolated studies that lack comprehensive clinical application to direct therapy. As a result, there is a wide discrepancy in the current treatment algorithm based on institution and provider, ultimately likely due to this large heterogeneity in the literature regarding reported outcomes of breast reconstruction in the setting of PMRT [12,18,19,31,32,33,34,35].

To date, only 2 studies have been found to compare patient outcomes at a single center based on the timing of PMRT [38,39]. One of these studies compared 49 patients who underwent exchange with a permanent implant within six months of their completion of PMRT (average, 3.4 months) to 39 patients who underwent exchange later than six months post-PMRT (average, 8.6 months) [39]. Expectedly, those patients who underwent an extended interval to exchange following PMRT experienced a lower incidence of failed reconstruction (7.7% vs. 22.4%; *p* = 0.04) [39]. The other study found similar results comparing patients who underwent an exchange prior to or after four months. The authors observed a decreased rate of complications in the group who underwent the later exchange; however, this was not shown to be statistically significant [38]. Overall, the findings from these two studies suggest that allowing for more time to pass between PMRT and exchange to a permanent prosthesis could potentially increase the likelihood of a successful reconstruction.

Limitations of this systematic review of the literature on PMRT before and after IBR include a minimal number of articles reporting outcomes specifically relevant to complications following PMRT before or after implant placement. While several articles present such data, they do not specifically state the infection rate or rate of explantation. Furthermore, patient comorbidity was not adequately reported, which may have affected the results of our analysis. For example, we cannot rule out the possibility that the associations found in this study are due to confounding variables between patient subgroups, rather than to the timing variance in PMRT treatment. Furthermore, we recognize that over the past 20 years, the use of biomaterials, implant technology, and movement from sub-muscular, dual-plane, and pre-pectoral, has represented a substantial shift in breast reconstruction techniques and outcomes, making extensive data analysis difficult with this study design. Finally, there is an apparent disparity in the balance of studies analyzed between PMRT prior to implant placement (n = 10), and PMRT after implant placement (n = 4); while the overall number of patients analyzed in each group was sufficient to achieve statistical significance in our chi-square analysis, we recognize that the paucity of studies addressing outcomes in PMRT after implant placement may confound the results of this analysis.

For example, there are some studies which, by virtue of the low number of reported cases of PMRT patients, reported atypically higher rates of infection, which may have impacted the range of the average rates analyzed. The results of this statistical analysis were thoroughly reviewed by our institution’s biostatistics division, and although there are limitations to this study, the statistical methods carried out in the study design were accurate for the presented data. Thus, while the findings illustrated in our analysis are encouraging, further in-depth studies are needed to validate the relevance of these associations.

## 5. Conclusions

Despite the higher incidence of complications, implant-based reconstruction remains the preferred form of reconstruction offered to women who receive PMRT, thus preserving the option of delayed autologous reconstruction. This study suggests that patients receiving PMRT before permanent implant placement in 2-stage expander–implant reconstruction, may have a higher risk of developing an infection. There was no difference in the rate of explantation between the two subgroups.

## Figures and Tables

**Table 1 medicina-55-00226-t001:** Summary of articles collected from the literature on postmastectomy radiation therapy (PMRT) prior to and following implant-based breast reconstruction (IBR); pts = patients; F/U = follow-up (months).

PMRT Before Implant Placement									
Author	Year	N pts	N Breasts	Age, y (mean)	F/U (mo)	Pain	Wound Dehiscence	Infection	Explantation
Nava et al.	2011	50	50	49	-	-	-	10 (20%)	20 (40%)
Baschnagel et al.	2012	90	90	45	24.1	1 (1.1%)	4 (4.4%)	7 (7.8%)	3 (3.3%)
Brooks et al.	2012	-	97	48.5	40.8	-	8 (8.2%)	10 (10.3%)	16 (16.5%)
Sbitany et al.	2014	-	113	43.9	25	-	2 (6.1%)	62 (54.9%)	20 (17.7%)
Ho et al.	2014	-	113	46.1	37.3	-	4 (3.5%)	4 (3.4%)	15 (13.3%)
Hirsch et al.	2014	237	240	47	33	9 (3.8%)		26 (10.8%)	33 (14%)
Fowble et al.	2015	99	99	44	45	-	-	-	18 (18%)
Santosa et al.	2016	104	176	47.7	16.8	-	5 (2.8%)	14 (8.0%)	3 (1.7%)
Chen et al.	2016	38	38	50.3	37.8	-	5 (13.2%)	4 (10.5%)	1 (2.6%)
Wang et al.	2016	-	129	47	26	-	29 (22%)	83 (64%)	19 (15%)
Pooled totals			1145						
**PMRT After Implant Placement**									
**Author**	**Year**	**N pts**	**N Breasts**	**Age, y (mean)**	**F/U (mo)**	**Pain**	**Wound Dehiscence**	**Infection**	**Explantation**
Anderson et al.	2009	74	84	49	48	-	-	-	3 (3.6%)
Nava et al.	2011	109	109	49	-	-	-	-	7 (6.4%)
Ho et al.	2012	151	151	44	86	-	-	14 (9.3%)	38 (25.2%)
Santosa et al.	2016	46	76	45	14.1	-	0 (0%)	8 (10.5%)	0 (0%)
Pooled totals			420						

**Table 2 medicina-55-00226-t002:** Comparison of respective rates of infection between subgroups: (1) PMRT before vs. (2) PMRT after implant placement in 2-stage expander-implant breast reconstruction; pts = patients; F/U = follow-up (months).

PMRT Before Implant Placement							
Author	Year	N pts	N Breasts	Age, y (mean)	F/U (mo)	Infection	
Nava et al.	2011	50	50	49	-	10 (20%)	
Baschnagel et al.	2012	90	90	45	24.1	7 (7.8%)	
Brooks et al.	2012	-	97	48.5	40.8	10 (10.3%)	
Sbitany et al.	2014	-	113	43.9	25	62 (54.9%)	
Ho et al.	2014	-	113	46.1	37.3	4 (3.4%)	
Hirsch et al.	2014	237	240	47	33	26 (10.8%)	
Santosa et al.	2016	104	176	47.7	16.8	14 (8.0%)	
Chen et al.	2016	38	38	50.3	37.8	4 (10.5%)	
Wang et al.	2016	-	129	47	26	83 (64%)	
Pooled totals			1046			220 (21.03%)	* *p* = 0.000079
							significant at *p* < 0.05
**PMRT After Implant Placement**							Chi-square value = 15.5813
**Author**	**Year**	**N pts**	**N Breasts**	**Age, y (mean)**	**F/U (mo)**	**Infection**	
Ho et al.	2012	151	151	44	86	14 (9.3%)	
Santosa et al.	2016	46	76	45	14.1	8 (10.5%)	
Pooled totals			227			22 (9.69%)	

**Table 3 medicina-55-00226-t003:** Comparison of respective rates of explantation between sub-groups: (1) PMRT before vs. (2) PMRT after implant placement in two-stage expander-implant breast reconstruction; pts = patients; F/U = follow-up (months).

PMRT Before Implant Placement
Author	Year	N pts	N Breasts	Age, y (mean)	F/U (mo)	Explantation	
Nava et al.	2011	50	50	49	-	20 (40%)	
Baschnagel et al.	2012	90	90	45	24.1	3 (3.3%)	
Brooks et al.	2012	-	97	48.5	40.8	16 (16.5%)	
Sbitany et al.	2014	-	113	43.9	25	20 (17.7%)	
Ho et al.	2014	-	113	46.1	37.3	15 (13.3%)	
Hirsch et al.	2014	237	240	47	33	33 (14%)	
Fowble et al.	2015	99	99	44	45	18 (18%)	
Santosa et al.	2016	104	176	47.7	16.8	3 (1.7%)	
Chen et al.	2016	38	38	50.3	37.8	1 (2.6%)	
Wang et al.	2016	-	129	47	26	19 (15%)	
Pooled totals			1145			148 (12.93%)	** p* = 0.427821
							Chi-square value = 0.6287
**PMRT After Implant Placement**
**Author**	**Year**	**N pts**	**N Breasts**	**Age, y (mean)**	**F/U (mo)**	**Explantation**	
Anderson et al.	2009	74	84	49	48	3 (3.6%)	
Nava et al.	2011	109	109	49	-	7 (6.4%)	
Ho et al.	2012	151	151	44	86	38 (25.2%)	
Santosa et al.	2016	46	76	45	14.1	0 (0%)	
Pooled totals			420			48 (11.43%)

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
