# Peer review of "Postmastectomy Radiation Therapy (PMRT) before and after 2-Stage Expander-Implant Breast Reconstruction: A Systematic Review"

_medicina, 2019, doi:10.3390/medicina55060226_

Round 1
Reviewer 1 Report
This is a review on reconstruction complications comparing PMRT before vs after implant placement. I enjoyed reading this review, however, I have some major concerns that should be addressed by the authors.
The comparison is not well balanced (10 studies for PMRT before implant and 4 studies for PMRT after implant. A comment in the discussion should be made regarding this imbalance.
In the PMRT before implant studies, 8/10 studies had an infection rate ranging between 7.8-20%. Two studies had an infection rate of 64% and 54.9%. These 2 studies showed unusual high rate of infection and they skewed the average. These 2 studies should be analyzed in more details and the readers need to know if there are any reason why the majority of their patients presented with infection.
I expect that these 2 issues be addressed in the discussion. perhaps a statistician should look at the analysis part.
Author Response
Dear Reviewer,
Thank you very much for your thorough review of our analysis! We certainly agree with your recommendations and have addressed each point you cite in your comments in the following paragraph, which has been added to our Discussion section. Furthermore, we have addressed the statistical methods with our institution's biostatistics division (prior to submission of this article), and have made this formal statement in the same paragraph below:
"Finally, there is an apparent disparity in the balance of studies analyzed between PMRT prior to implant placement (n=10) and PMRT after implant placement (n=4); while the overall number of patients analyzed in each group was sufficient to achieve statistical significance in our chi-square analysis, we recognize that the paucity of studies addressing outcomes in PMRT after implant placement may confound the results of this analysis. For example, there are some studies which, by virtue of the low number of reported cases of PMRT patients, reported atypically higher rates of infection, which may have impacted the range of the average rates analyzed. The results of this statistical analysis were thoroughly reviewed by our institution’s biostatistics division, and although there are limitations to this study, the statistical methods carried out in the study design were accurate for the presented data."
Thank you again very much for your review. We look forward to hearing from the Editorial Board in regards to this revised, improved manuscript.
Reviewer 2 Report
Dear Authors,
The study is interesting, but it is not a meta-analysis. It does not comply with meta-analysis guidelines (PRISMA), there is no proper meta-analytical analysis, the statistics performed are not clear.
The article should be either rewritten as a systematic (or non-systematic) review, or as a proper meta-analysis
Author Response
Thank you very much for your thorough review of our manuscript! We completely agree with your comments. This study is much better defined as a Systematic Review rather than a Meta-Analysis; thus, we have addressed this in the manuscript text, rephrased the Title, and have addressed the shortcomings of the analysis in the Discussion section to explain why meta-analytic methods were not amenable in this particular analysis.
Thank you again for your review. We look forward to hearing from the Editorial Board in regards to our revised, improved manuscript for publication consideration.
Round 2
Reviewer 2 Report
The authors have complied with the comments of the reviewers. As such, the article is acceptable to be published.